# Theoretical Study on the Structural, Elastic, Electronic and Thermodynamic Properties of Long-Period Superstructures h- and r-Al_2_Ti under High Pressure

**DOI:** 10.3390/ma15124236

**Published:** 2022-06-15

**Authors:** Yufeng Wen, Xianshi Zeng, Yuanxiu Ye, Qingdong Gou, Bo Liu, Zhangli Lai, Daguo Jiang, Xinyuan Sun, Minghui Wu

**Affiliations:** 1School of Mathematical Sciences and Physics, Jinggangshan University, Ji’an 343009, China; wenyufeng@jgsu.edu.cn (Y.W.); zengxueliang@163.com (X.Z.); jgsxy_yyx@sohu.com (Y.Y.); gouqingdong@163.com (Q.G.); liubo8203@126.com (B.L.); 15979627768@163.com (Z.L.); sxy5306@126.com (X.S.); 2Fujian Key Laboratory of Functional Marine Sensing Materials, College of Materials and Chemical Engineering, Minjiang University, Fuzhou 350108, China; minghuiwu@mju.edu.cn

**Keywords:** first-principles calculations, high pressure, structural properties, elastic properties, electronic structures, thermodynamic properties, long-period superstructures

## Abstract

The formations of long-period superstructures strongly influence the properties of Al-rich L1${}_{0}$-TiAl intermetallic alloys. To soundly understand the role of the superstructures in the alloys, fundamentals about them have to be known. In the present work, the structural, elastic, electronic and thermodynamic properties of h- and r-Al${}_{2}$Ti long-period superstructures under pressure up to 30 GPa were systematically investigated using first-principles calculations based on density functional theory. The pressure dependence of structural parameters, single-crystal elastic constants, polycrystalline elastic modulus, Cauchy pressures and elastic anisotropy were successfully calculated and discussed. The total and partial densities of states at different pressures were also successfully calculated and discussed. Furthermore, combining with quasi-harmonic approximation, the effects of the pressure on the temperature dependent volume, isothermal bulk modulus, thermal expansion coefficient, heat capacity and Gibbs free energy difference were successfully obtained and discussed. Our results were consistent with the available experimental and theoretical values.

## 1. Introduction

Intermetallic alloys based on the L1${}_{0}$-TiAl compound are one of the few classes of emerging materials that have the potential to be used in demanding high-temperature structural applications whenever specific strength and stiffness are of major concern because they offer an attractive combination of low density and good oxidation and ignition resistance with unique mechanical properties [1]. In the Ti-Al phase diagram, the L1${}_{0}$-TiAl phase can exist in a wide composition range from the stoichiometry to the Al-rich side [2]. However, the formations of long-period superstructures (LPSs) in the L1${}_{0}$-TiAl matrix were often observed experimentally in Al-rich intermetallic alloys [3,4,5,6,7,8,9,10,11,12,13,14,15,16,17,18,19,20]. The occurrence of such superstructures can produce a strong impact on the operative deformation mode [12], dislocation configuration [13], anomalous strengthening [14], and mechanical properties [10,11,12,13,14,15,16,17,18,19,20].

In Al-rich L1${}_{0}$-TiAl intermetallic alloys, the crystal structures and microstructures of the LPSs are very sensitive to the alloy composition and heat treatment conditions [3,4,5,6,7,8,9,11,13]. The LPSs with constant composition were distinguished as Al${}_{5}$Ti${}_{3}$, Al${}_{2}$Ti, Al${}_{11}$Ti${}_{5}$, Al${}_{5}$Ti${}_{2}$, and Al${}_{3}$Ti [4]. They have a similar unit cell with different periodicity based on the L1${}_{0}$ lattice. In one of these stoichiometric LPSs, there are two types of crystal lattice in the Al${}_{2}$Ti superstructure. One is an orthorhombic lattice, called h-Al${}_{2}$Ti, and the other is a tetragonal lattice, called r-Al${}_{2}$Ti [21,22]. The h-Al${}_{2}$Ti LPS was observed experimentally to transform into the r-Al${}_{2}$Ti one [4,11]. However, the phase transition temperature and the thermal stability of h-Al${}_{2}$Ti still remains unclear and somewhat controversial to date.

Although quite a few experiments were reported with the aim of understanding the thermal stabilities and phase transformation of h- and r-Al${}_{2}$Ti LPSs in Al-rich L1${}_{0}$-TiAl alloys, theoretical calculations on the Al${}_{2}$Ti ones are still absent. The r-Al${}_{2}$Ti LPS was first calculated by Watson and Weinert employing full-potential linearized augmented Slater-type orbital [23]. Subsequently, the structural energetics of h- and r-Al${}_{2}$Ti LPSs are calculated by Ghosh and Asta using the first-principles method based on density-functional theory (DFT) [24]. Recently, first-principles calculations based on DFT was taken by Tang et al. to study the structural, electronic and elastic properties of both Al${}_{2}$Ti LPSs at ground-state [25]. Most recently, the DFT-based first-principles calculations were also taken by Ghosh et al. to investigate the structural stability of both Al${}_{2}$Ti LPSs at ground-state [26].

It is known that the electronic, elastic and thermodynamic properties of intermetallic compounds under pressure are vital to the design and development of novel materials for structural applications. These properties are determined by the crystal structures. To the best of our knowledge, there is a real lack of knowledge on the structural, electronic, elastic and thermodynamic properties of h- and r-Al${}_{2}$Ti LPSs under high pressure up to now. This lack has prompted us to investigate them. This work aims to present a systematical study of the structural, electronic, elastic and thermodynamic properties of both Al${}_{2}$Ti LPSs under high pressure, using the DFT-based first-principles method in combination with quasi-harmonic approximation (QHA). The modelling and theoretical methods used in this work are described in Section 2. The results are discussed in Section 3. Finally, the conclusions of the work are drawn in Section 4.

## 2. Modelling and Methods

### 2.1. Theoretical Models

The crystallographic data of the LPSs h- and r-Al${}_{2}$Ti have been determined experimentally by X-ray diffraction [21,22]. The h LPS belongs to an orthorhombic structure with the space group of Cmmm, while the r LPS has a tetragonal structure with the space group of I41/amd. In the h LPS, three crystallographically inequivalent Al atoms are located on the 2a, 2c and 4h Wyckoff sites with a Ti atom on the 4g site, while in the r LPS, two Al and one Ti atoms sit in the 8e site. The inital theoretical models of both LPSs are built according to the crystallographic information from Refs. [21,22], as shown in Figure 1.

### 2.2. Computational Details

The DFT-based first-principles calculations were performed using the projector augmented wave (PAW) method and a plane wave basis set [27,28], as implemented in Vienna Ab initio Simulation Package (VASP) [29,30,31]. The exchange-correlation functional were treated by the generalized gradient approximation (GGA) formulated by Perdew–Burke–Ernzerhof (PBE) [32]. The configurations Ti 3s${}^{2}$3p${}^{6}$3d${}^{2}$4s${}^{2}$ and Al 3s${}^{2}$3p${}^{1}$ were treated as valence electrons. A cutoff energy of 600 eV was specified for the plane wave set. A global break condition of $10^{-6}$ eV/atom was specified for the electronic self-consistency loop. The 4×13×12 and 13×13×2 Monkhorst–Pack methods [33] were used as the Brillouin-zone sampling for the h and r LPSs, respectively. At a given pressure, the unit cells of both LPSs were fully relaxed with respect to the volume, shape and internal atomic positions until the atomic forces of less than 0.01 eV/Å. The calculations of total energy and density of states (DOS) were performed using the tetrahedron method with the Blöchl corrections [34].

### 2.3. Elastic Properties

The elastic properties of crystals are essential for a sound understanding of their fundamental physical properties and give important information concerning the nature of the forces operating in crystals. In particular, they provide valuable information on the mechanical stability, stiffness, ductility/brittleness behavior, strength, hardness, and bonding characteristic between adjacent atomic planes and anisotropic character of the bonding. For an orthorhombic crystal, there are nine independent single-crystal elastic constants, i.e., $C_{11}$, $C_{12}$, $C_{13}$, $C_{22}$,$C_{23}$, $C_{33}$, $C_{44}$, $C_{55}$, $C_{66}$. As a special case of the orthorhombic system, these constants of the tetragonal system are reduced to six independent components $C_{11}$($=C_{22}$), $C_{12}$, $C_{13}$($=C_{23}$), $C_{33}$, $C_{44}$($=C_{55}$), $C_{66}$. The strain-stress relationship [35] was employed to calculate the elastic constants of both LPSs, as implemented in the VASP. The elastic tensor is determined by performing six finite distortions of the lattice and deriving the elastic constants from the strain-stress relationship. The final elastic constants include both the contributions for distortions with rigid ions and the contributions from the ionic relaxations [36]. At a given pressure, the calculations of elastic constants were conducted on the basis of the optimized structural parameters at the pressure.

The polycrystalline elastic moduli of isotropic materials can be determined from the single crystal elastic constants by the Voigt–Reuss–Hill (VRH) approximation [37]. For an orthorhombic crystal, the bulk and shear moduli in VRH approximation are as follows: (1)$$\begin{array}{c} B_{V}=\left(C_{11}+C_{22}+C_{33}+2C_{12}+2C_{13}+2C_{23}\right)/9,\\ G_{V}=\left(C_{11}+C_{22}+C_{33}+3C_{44}+3C_{55}+3C_{66}-C_{12}-C_{13}-C_{23}\right)/15,\\ B_{R}=\Delta /[C_{11}\left(C_{22}+C_{33}-2C_{23}\right)+C_{22}\left(C_{33}-2C_{13}\right)-2C_{33}C_{12}\\ \;\;\;\;+C_{12}\left(2C_{23}-C_{12}\right)+C_{13}\left(2C_{12}-C_{13}\right)+C_{23}\left(2C_{13}-C_{23}\right)],\\ G_{R}=15/\{4[C_{11}\left(C_{22}+C_{33}+C_{23}\right)+C_{22}\left(C_{33}+C_{13}\right)+C_{33}C_{12}\\ \;\;\;\;-C_{12}\left(C_{23}+C_{12}\right)-C_{13}\left(C_{12}+C_{13}\right)-C_{23}\left(C_{13}+C_{23}\right)]/\Delta \\ +3/C_{44}+3/C_{55}+3/C_{66}\},\\ B_{H}=\left(B_{V}+B_{R}\right)/2,\;\;G_{H}=\left(G_{V}+G_{R}\right)/2,\\ \Delta =C_{13}\left(C_{12}C_{23}-C_{13}C_{22}\right)+C_{23}\left(C_{12}C_{13}-C_{23}C_{11}\right)+C_{33}\left(C_{11}C_{22}-C_{12}^{2}\right). \end{array}$$
where, $B_{V}$, $B_{R}$ and $B_{H}$ as well as $G_{V}$,$G_{R}$ and $G_{H}$ represent the Voigt, Reuss and Hill bulk as well as shear moduli, respectively. Then the Young’s modulus $E_{H}$ and Poisson’s ratio $\nu_{H}$ are given in terms of $B_{H}$ and $G_{H}$ by [37]
(2)$$\begin{array}{c} E_{H}=9B_{H}G_{H}/\left(3B_{H}+G_{H}\right),\\ \nu_{H}=\left(3B_{H}-2G_{H}\right)/\left(6B_{H}+2G_{H}\right). \end{array}$$

Elastic anisotropy plays an important role in diverse applications of crystalline materials such as the mechanical properties of nickel-based superalloys, microscale cracking in ceramics, phase transformations, dislocation dynamics, development of plastic deformation, enhanced positively charged defect mobility, alignment or misalignment of quantum dots, texture in nanoscale shape-memory alloys, and plastic relaxation in thin-film metallics [38,39]. In terms of elastic compliances $S_{ij}$($=C_{ij}^{-1}$), the variation of bulk modulus with direction for an orthorhombic crystal can be calculated by [40]
(3)$$\begin{array}{c} B^{-1}=\left(S_{11}+S_{12}+S_{13}\right)\alpha^{2}+\left(S_{12}+S_{22}+S_{23}\right)\beta^{2}+\left(S_{13}+S_{23}+S_{33}\right)\gamma^{2}, \end{array}$$
where, α, β and γ are direction cosines. Meanwhile, the linear bulk moduli along the orthogonal axes ($B_{a}$, $B_{b}$, $B_{c}$) can be calculated by [41]
(4)$$\begin{array}{c} B_{a}=a\frac{dP}{da}=\frac{\Lambda }{1+\xi +\chi^{\prime }}\\ B_{b}=b\frac{dP}{db}=\frac{B_{a}}{\xi^{\prime }}\\ B_{c}=a\frac{dP}{dc}=\frac{B_{a}}{\chi^{\prime }}\\ \Lambda =C_{11}+2C_{12}\xi +C+22\xi^{2}+2C_{13}\chi +C_{33}\chi^{2}+2C_{23}\xi \chi ,\\ \xi =\frac{\left(C_{11}-C_{12}\right)\left(C_{33}-C_{13}\right)-\left(C_{23}-C_{13}\right)\left(C_{11}-C_{13}\right)}{\left(C_{33}-C_{13}\right)\left(C_{22}-C_{12}\right)-\left(C_{13}-C_{23}\right)\left(C_{12}-C_{23}\right)},\\ \chi =\frac{\left(C_{22}-C_{12}\right)\left(C_{11}-C_{13}\right)-\left(C_{11}-C_{12}\right)\left(C_{23}-C_{12}\right)}{\left(C_{22}-C_{12}\right)\left(C_{33}-C_{13}\right)-\left(C_{12}-C_{23}\right)\left(C_{13}-C_{23}\right)}. \end{array}$$

In addition to bulk modulus, the directional dependence of shear modulus *G* is important for understanding the elastic anisotropy for crystalline materials. For an orthorhombic crystal, the shear modulus in any orientation can be calculated by [40]
(5)$$\begin{array}{c} G^{-1}=4S_{11}\alpha_{1}^{2}\alpha_{2}^{2}+4S_{22}\beta_{1}^{2}\beta_{2}^{2}+4S_{33}\gamma_{1}^{2}\gamma_{2}^{2}+8S_{12}\alpha_{1}\alpha_{2}\beta_{1}\beta_{2}+8S_{23}\beta_{1}\beta_{2}\gamma_{1}\gamma_{2}+8S_{13}\alpha_{1}\alpha_{2}\gamma_{1}\gamma_{2}\\ +S_{44}{\left(\beta_{1}\gamma_{2}+\beta_{2}\gamma_{1}\right)}^{2}+S_{55}{\left(\alpha_{1}\gamma_{2}+\alpha_{2}\gamma_{1}\right)}^{2}+S_{66}{\left(\gamma_{1}\beta_{2}+\alpha_{2}\beta_{1}\right)}^{2}. \end{array}$$
where, $\alpha_{1}$, $\beta_{1}$ and $\gamma_{1}$ are direction cosines of the shear stress direction [uvw], $\alpha_{2}$, $\beta_{2}$ and $\gamma_{2}$ are direction cosines of the shear plane normal [HKL]. Meanwhile, the shear anisotropic factors for the {100} planes between the <011> and <010> directions, the {010} planes between the <101> and <001> directions, and the {001} planes between the <110> and <010> directions can be calculated correspondingly by [42]
(6)$$\begin{array}{c} A_{\left\{100\right\}}=\frac{4C_{44}}{C_{11}+C_{33}-2C_{13}},\\ A_{\left\{010\right\}}=\frac{4C_{55}}{C_{22}+C_{33}-2C_{23}},\\ A_{\left\{001\right\}}=\frac{4C_{66}}{C_{11}+C_{22}-2C_{12}}. \end{array}$$

Besides, the directional dependence of Young’s modulus *E* is also important for understanding the elastic anisotropy for crystalline materials. The Young’s modulus in any orientation for an orthorhombic crystal can be calculated by [40]
(7)$$\begin{array}{c} E^{-1}=S_{11}\alpha^{4}+S_{22}\beta^{4}+S_{33}\gamma^{4}+\left(2S_{12}+S_{66}\right)\alpha^{2}\beta^{2}+\left(2S_{23}+S_{44}\right)\beta^{2}\gamma^{2}+\left(2S_{13}+S_{55}\right)\alpha^{2}\gamma^{2}. \end{array}$$

Furthermore, the universal ($A^{U}$) and log-Euclidean ($A^{L}$) anisotropy indexes can be calculated by [38,39]
(8)$$\begin{array}{c} A^{U}=\frac{B_{V}}{B_{R}}+5\frac{G_{V}}{G_{R}}-6,\\ A^{L}=\sqrt{{\left[ln\left(\frac{B_{V}}{B_{R}}\right)\right]}^{2}+5{\left[ln\left(\frac{G_{V}}{G_{R}}\right)\right]}^{2}}. \end{array}$$

### 2.4. Thermodynamic Properties

Thermodynamic properties were calculated using the VASP coupled with the phonopy package under the QHA [43]. After the supercells with atomic displacements were created from a unit cell via the phonopy package, the VASP calculations for the finite displacement method were undertaken based on them to obtain force constants. The supercell contains 1×3×3 and 3×3×1 unit cells for the h and r LPSs, respectively. The Brillouin zone was sampled using 3×3×3 and 3×3×2 Gamma centered Monkhorst–Pack grids for the h and r LPS, respectively. A part of the dynamical matrix was built from the force constants. Phonon frequencies and eigenvectors were computed from the dynamical matrices. Then, the phonon related properties were calculated from phonon frequencies and eigenvectors via the phonopy package.

## 3. Results and Discussion

### 3.1. Ground-State Bulk Properties

Our calculated ground-state bulk properties of the LPSs h- and r-Al${}_{2}$Ti are listed in Table 1. These bulk properties include structural parameters $a_{0}$, $b_{0}$, $c_{0}$ and $V_{0}$, single-crystal elastic constants $C_{ij}$ s, and polycrystalline elastic moduli $B_{H}$, $G_{H}$, $E_{H}$ and $\nu_{H}$. Meanwhile, the experimental and other theoretical results are also listed in the table for comparison. For the structural parameters, the relative error between our theoretical value and the experimental data for the h(r) LPS is 0.53 (0.05)% for the lattice constant $a_{0}$, 0.68 (0.05)% for the $b_{0}$, 0.77 (0.01)% for the $c_{0}$ and 0.77 (0.09)% for the unit cell volume $V_{0}$ [21,22]. For both LPSs, the present values of these structural parameters are in excellent accordance with the previous theoretical results [24,25,26,44]. For the elastic constants and moduli, the present values are also in good agreement with the previous theoretical results [25,44].

### 3.2. Structural Properties

The optimized lattice constants and unit cell volumes of the LPSs h- and r-Al${}_{2}$Ti under pressure up to 30 GPa at 0 K are listed in Table 2. Various structural parameters of both LPSs at 0 K decrease monotonically with increasing pressure. The relative changes of these parameters and the ratios of lattice parameters for both LPSs at 0 K as a function of pressure are plotted in Figure 2. One can see from Figure 2a,b that these relative changes gradually reduce as the pressues increases, and the relative change of the volume decreases more rapidly than those of the lattice constants. Meanwhile, the decrease of the $b/b_{0}$ for the h LPS is slightly faster than that of its $a/a_{0}$ and slightly slower than that of its $c/c_{0}$, whereas the decrease of the $a/a_{0}$ for the r LPS is slightly faster than that of the counterpart $c/c_{0}$. These show that the incompressibility of the *b* axis is slightly weaker than that of the *a* axis and slightly stronger than that of the *c* axis for the h LPS, while the incompressibility of the *a* axis is slightly weaker than that of the *c* one for the r LPS. One can also see from Figure 2c that the ratios of a/b and c/b for the h LPS and c/a for the r one are all almost unchanged with increasing the pressure, meaning that both LPSs have a pressure isotropic structure.

### 3.3. Elastic Properties

The present elastic constants of both LPSs under pressure up to 30 GPa at 0 K are listed in Table 3. The elastic constants $C_{11}$, $C_{22}$, $C_{33}$ represent the elasticity in length, and $C_{12}$, $C_{13}$, $C_{23}$, $C_{44}$, $C_{55}$, $C_{66}$ are related to the elasticity in shape. At zero temperature, all of the constants increase monotonically with increasing the pressure. The values of $C_{11}$, $C_{22}$ and $C_{33}$ for both LPSs are always greater than those of the corresponding $C_{12}$, $C_{13}$, $C_{23}$, $C_{44}$, $C_{55}$ and $C_{66}$ in the pressure range from 0 to 30 GPa, indicating the higher possibility of occurrence for the changes in shape than those in length for both LPSs at a given pressure. The values of $C_{11}$, $C_{22}$ and $C_{33}$ for the h LPS are always in the order of $C_{11}>C_{22}>C_{33}$ in the pressure range from 0 to 30 GPa, implying that the incompressibility along *a* axis is the strongest and that along *c* axis is the weakest. However, the value of $C_{11}$ for the r LPS is always smaller than that of $C_{33}$ in the pressure range from 0 to 30 GPa, indicating the weaker incompressibility along the *a* axis than along the *c* axis. These results are consistent with the analysis on the compressions of lattice constants for both LPSs. Additionally, the values of $C_{44}$, $C_{55}$ and $C_{66}$ for the h LPS are always in the order of $C_{44}\approx C_{55}>C_{66}$ in the pressure range from 0 to 30 GPa, indicating that its resistance to shear deformation is the weakest on {001} planes. However, the value of $C_{44}$ for the r LPS is always smaller than that of $C_{66}$ in the pressure range from 0 to 30 GPa, implying that its resistance to shear deformation is the strongest on {001} planes.

Various elastic constants of both LPSs at 0 K as a function of pressure are plotted in Figure 3. Clearly, each elastic constant increases rapidly with increasing the pressure, and the constants $C_{11}$, $C_{22}$, $C_{33}$ are relatively more sensitive to pressure than the other constants. All of the elastic constants exhibit a linear growth trend with increasing the pressure. For the h LPS, the linear relationships between the elastic constants $C_{ij}$ and the pressure *P* are as follows:(9)$$\begin{array}{c} C_{11}=213.96434+4.37139P\;\left(R^{2}=0.99719\right),\\ C_{12}=57.24268+1.87076P\;\left(R^{2}=0.99842\right),\\ C_{13}=60.33057+1.78942P\;\left(R^{2}=0.99907\right),\\ C_{22}=204.41655+3.93892P\;\left(R^{2}=0.9973\right),\\ C_{23}=68.93576+2.08766P\;\left(R^{2}=0.99839\right),\\ C_{33}=195.62559+3.73204P\;\left(R^{2}=0.99683\right),\\ C_{44}=99.48011+2.40161P\;\left(R^{2}=0.99748\right),\\ C_{55}=96.37195+2.51475P\;\left(R^{2}=0.99829\right),\\ C_{66}=84.29467+2.04193P\;\left(R^{2}=0.99704\right), \end{array}$$
and for the r LPS they are as follows: (10)$$\begin{array}{c} C_{11}=202.56406+3.78756P\;\left(R^{2}=0.99598\right),\\ C_{12}=68.71253+2.03245P\;\left(R^{2}=0.99794\right),\\ C_{13}=58.44692+1.86842P\;\left(R^{2}=0.99885\right),\\ C_{33}=213.13705+4.39402P\;\left(R^{2}=0.99776\right),\\ C_{44}=88.6517+2.23467P\;\left(R^{2}=0.99754\right),\\ C_{66}=97.614+2.48732P\;\left(R^{2}=0.9984\right). \end{array}$$
where, $R^{2}$ is the coefficient of determination, and a value of 1.0 for $R^{2}$ indicates a perfect positive linear relationship.

For an orthorhombic system under hydrostatic pressure, the necessary and sufficient Born criteria are as follows [45,46]: (11)$$\begin{array}{c} {\tilde{C}}_{11}>0,\;\;{\tilde{C}}_{11}{\tilde{C}}_{22}>{\tilde{C}}_{12}^{2},\\ {\tilde{C}}_{11}{\tilde{C}}_{22}{\tilde{C}}_{33}+2{\tilde{C}}_{12}{\tilde{C}}_{13}{\tilde{C}}_{23}-{\tilde{C}}_{11}{\tilde{C}}_{23}^{2}-{\tilde{C}}_{22}{\tilde{C}}_{13}^{2}-{\tilde{C}}_{33}{\tilde{C}}_{12}^{2}>0,\\ {\tilde{C}}_{44}>0,\;\;{\tilde{C}}_{55}>0,\;\;{\tilde{C}}_{66}>0,\\ {\tilde{C}}_{ii}=C_{ii}-P\left(i=1∼6\right),\;{\tilde{C}}_{12}=C_{12}+P,\;{\tilde{C}}_{13}=C_{13}+P,\;{\tilde{C}}_{23}=C_{23}+P. \end{array}$$

In terms of the values of $C_{ij}$ given in Table 3, it is concluded that both h- and r-Al${}_{2}$Ti can satisfy the above criteria completely in the pressure range from 0 to 30 GPa. The calculated phonon dispersions of both LPSs at 0 GPa and 30 GPa are shown in Figure 4 to further verify their stabilities. No imaginary frequency is observed in the phonon spectra of both LPSs, indicating their dynamical stabilities at 0 GPa and 30 GPa.

The present elastic moduli of both LPSs under pressure up to 30 GPa at 0 K are listed in Table 4. At zero temperature, the moduli $B_{H}$, $G_{H}$, $E_{H}$, and the ratios $B_{H}/G_{H}$, $\nu_{H}$ increase monotonically with increasing the pressure. As suggested by Pugh [47], the $B_{H}/G_{H}$ ratio can be used to distinguish the ductility or brittleness of polycrystalline materials. A higher (lower) $B_{H}/G_{H}$ value is related to ductility (brittleness) of a material. The critical value which separates ductility and brittleness is about 1.75. From Table 4, one can see that the obtained $B_{H}/G_{H}$ ratios of both LPSs at 0 K are significantly smaller than 1.75 in the pressure range from 0 to 30 GPa. Thus, at zero temperature both LPSs have brittle features in the pressure range from 0 to 30 GPa. At identical pressures, the $B_{H}/G_{H}$ value of the h LPS is always lower than that of the r LPS. So, it can be concluded that the h LPS is always more brittle than the r one in the pressure range from 0 to 30 GPa.

As suggested by Frantsevich [48], Poisson’s ratio can also be used to distinguish between ductility and brittleness. The smaller the Poisson’s ratio, the stronger the brittleness. The criticAl$\nu_{H}$ value of a material is 0.26. From Table 4, one can also see that the obtained Poisson’s ratios of both LPSs at 0 K are significantly smaller than 0.26 in the pressure range from 0 to 30 GPa. Meanwhile, the $\nu_{H}$ value of the h LPS at a given pressure is smaller than the corresponding one of the r LPS. Therefore, in the pressure range from 0 to 30 GPa, both LPSs at 0 K exhibit brittle behaviors, and the h LPs possesses a stronger brittleness as compared with the r one. These are consistent with the results of the $B_{H}/G_{H}$ ratios.

As suggested by Pettifor [49], Cauchy pressure for metals and intermetallics can describe the angular character of atomic bonding that relates to their brittle or ductile characteristics. For metallic bonding the Cauchy pressure is typically positive, while for directional bonding with angular character it is negative, with larger negative pressure representing a more directional character. In orthorhombic crystals, the Cauchy pressures are defined by $CP_{1}=C_{12}-C_{66}$, $CP_{2}=C_{13}-C_{55}$ and $CP_{3}=C_{23}-C_{44}$. The presently obtained Cauchy pressures of both LPSs under pressure up to 30 GPa at 0 K are also listed in Table 4. It can be seen from the table that all Cauchy pressures of both LPSs at 0 K are negative in the pressure range from 0 to 30 GPa, and they become larger negative with increasing the pressure. It is shown that at zero temperature both LPSs with more angular bonding become more brittle as the pressure increases, which agrees excellently with the results of the ratios $B_{H}/G_{H}$ and $\nu_{H}$.

The elastic moduli $B_{H}$, $G_{H}$, $E_{H}$ of both LPSs at 0 K as a function of pressure are plotted in Figure 5. Similar to the elastic constants, the values of $B_{H}$, $G_{H}$ and $E_{H}$ increase rapidly with increasing the pressure, and the modulus $E_{H}$ is relatively more sensitive to pressure than the other two. They exhibit a linear growth trend as the pressure increases. For the h LPS, the linear relationships between the moduli $B_{H}$, $G_{H}$, $E_{H}$ and the pressure *P* are as follows: (12)$$\begin{array}{c} B_{H}=109.66027+2.6138P\;\left(R^{2}=0.99796\right),\\ G_{H}=83.61274+1.70928P\;\left(R^{2}=0.99661\right),\\ E_{H}=200.03068+4.21929P\;\left(R^{2}=0.9969\right), \end{array}$$
and for the r LPS they are as follows: (13)$$\begin{array}{c} B_{H}=110.1696+2.61094P\;\left(R^{2}=0.99804\right),\\ G_{H}=83.10901+1.70764P\;\left(R^{2}=0.9964\right),\\ E_{H}=199.25306+4.21371P\;\left(R^{2}=0.99682\right). \end{array}$$

The present orientation dependence of bulk modulus for the LPSs h- and r-Al${}_{2}$Ti under pressure up to 30 GPa at 0 K are shown in Figure 6 and Figure 7, respectively. For these three-dimensional (3D) representation surfaces, the size of the bulk modulus is denoted by the length of the radius vector in arbitrary crystallographic directions and different colors. The 3D surface of an isotropic system must be a spherical shape with a color, and the nonsperical shape with different colors indicates the degree of anisotropy. Clearly, the directional dependences of *B* for both LPSs under any pressure at 0 K illustrate nonspherical 3D shapes with various colors, indicating their intrinsic anisotropies. The obtained maximal ($B_{max}$) and minimal ($B_{min}$) values and the $B_{max}/B_{min}$ ratio of bulk modulus for both LPSs under pressure up to 30 GPa at 0 K are listed in Table 5. One can find that the $B_{max}$ and $B_{min}$ values of both LPSs at 0 K increase gradually with increasing the pressure. The $B_{max}/B_{min}$ ratios of both LPSs, which are slightly greater than one in the studied entire pressure range, are also found to increase on the whole with increasing the pressure, indicating that their anisotropies are weak and increase with increasing the pressure. At identical pressures the $B_{max}/B_{min}$ value of the h LPS is always larger than that of the r LPS, showing a stronger elastic anisotropy of the h LPS than the r one in the pressure range from 0 to 30 GPa.

The present $B_{a}$, $B_{b}$, $B_{c}$ of both LPSs under different pressures at 0 K are also listed in Table 5. Comparing the linear bulk moduli with the corresponding extremum, the relations of $B_{a}=B_{max}$ and $B_{c}=B_{min}$ are found for the h LPS, while those of $B_{a}=B_{min}$ and $B_{c}=B_{max}$ are found for the r one. Meanwhile, the moduli along the orthogonal axes of both LPSs at 0 K increase monotonically with increasing the pressure. The change of the modulus along the *a* axis of the h LPS is the largest, followed by that along the *b* axis, and that along the *c* axis is the smallest. On the contrary, the change of the modulus along the *a* or *b* axis of the r LPS is smaller than that along the *c* axis. This means that at applied pressure the *a* axis of the h LPS is the most incompressible and its *c* axis is the least incompressible, while the *a* axis of the r LPS is more easily compressed than its *c* axis. To study the linear compressibility anisotropies of both LPSs, the anisotropic factors of linear bulk modulus along the *a* ($B_{Ba}=B_{a}/B_{b}$) and *c* ($B_{Bc}=B_{c}/B_{b}$) axes with respect to the *b* axis are further obtained and also listed in Table 5. If the factors have a value of one, the LPSs are isotropic. Otherwise, the factors have a larger departure from one, the LPSs have a stronger anisotropy. Similar to the $B_{max}/B_{min}$ ratios, the change trends of the $B_{Ba}$ and $B_{Bc}$ close to one for both LPSs at 0 K show the positive pressure effects on their weak linear compressibility anisotropies in the pressure region between 0 and 30 GPa.

The present orientation dependences of shear modulus for the LPSs h- and r-Al${}_{2}$Ti under pressure up to 30 GPa at 0 K are shown in Figure 8 and Figure 9, respectively. Clearly, all 3D shapes of the shear modulus for both LPSs at 0 K have a very strong nonsphericity, showing their very significant anisotropies. The obtained external values $G_{max}$, $G_{min}$ and the $G_{max}/G_{min}$ ratio of both LPSs under pressure up to 30 GPa at 0 K are listed in Table 6. It can be observed that the external values of both LPSs at 0 K increase gradually with increasing the pressure. The $G_{max}/G_{min}$ ratios significantly greater than one of both LPSs increase gradually with increasing the pressure in the pressure range from 0 to 30 GPa, showing the increase of their strong anisotropies with pressure. At identical pressures, the h LPS always has a larger $G_{max}/G_{min}$ value as compared with the r one, meaning that the anisotropy of the h LPS is always stronger than that of the r one over the entire studied pressure range.

The present shear anisotropic factors of both LPSs under pressure up to 30 GPa at 0 K are listed in Table 6. For these factors, a value of unity corresponds to isotropy, while any deviation from unity corresponds to an extent of anisotropy. Clearly, the values of the three factors for both LPSs are significantly greater than one in the pressure region between 0 and 30 GPa, and increase gradually with increasing the pressure. It is shown that in {100}, {010} and {001} planes, both LPSs possess strong shear anisotropies and the degrees of the anisotropies are increased with increasing the pressure. At a given pressure, the value of $A_{\left\{010\right\}}$ for the h LPS at 0 K is the largest, followed by $A_{\left\{100\right\}}$ and that of $A_{\left\{001\right\}}$ is the smallest. It is indicated that the anisotropy of the {010} shear planes between the <101> and <001> directions for the h LPS is always the strongest and that of its {001} shear planes between the <110> and <010> directions is always the weakest in the entire studied pressure range. However, the $A_{\left\{100\right\}}$ or $A_{\left\{010\right\}}$ value is smaller than the $A_{\left\{001\right\}}$ one for the r LPS at a given pressure and 0 K, showing the stronger anisotropy of its {001} shear planes between the <110> and <010> directions than its {100}({010}) shear planes between the <011>(<101>) and <010>(<001>) directions in the pressure range from 0 to 30 GPa. Comparing each shear anisotropic factor of both LPSs, it can be found that at identical pressures the h LPS has a larger $A_{\left\{100\right\}}$ or $A_{\left\{010\right\}}$ value, but a smaller $A_{\left\{001\right\}}$ value than the r one. Hence, over the entire studied pressure range, the h LPS has a stronger anisotropy of the {100}({010}) shear planes between the <011>(<101>) and <010>(<001>) directions, but a weaker anisotropy of the {001} shear planes between the <110> and <010> directions than the r one.

The present orientation dependences of Young’s modulus for the LPSs h- and r-Al${}_{2}$Ti under pressure up to 30 GPa at 0 K are shown in Figure 10 and Figure 11, respectively. Similar to the shear modulus above, all 3D shapes of Young’s modulus for both LPSs at 0 K also have a very strong nonsphericity, showing their very significant anisotropies. The obtained external values $E_{max}$, $E_{min}$ and the $E_{max}/E_{min}$ ratio of both LPSs under pressure up to 30 GPa at 0 K are listed in Table 7. The two extremes of the h and r LPSs at 0 K are increased gradually with increasing the pressure. Their $E_{max}/E_{min}$ ratios are significantly greater than one and increase gradually with increasing the pressure in the studied entire pressure range, showing the increase of their strong anisotropies with pressure. At identical pressures the $E_{max}/E_{min}$ value of the h LPS is always larger than that of the r one, showing the stronger anisotropy of the h LPS than the r one in the pressure range between 0 and 30 GPa.

The present $A^{U}$ and $A^{L}$ of both LPSs under pressure up to 30 GPa at 0 K are also listed in Table 7. For these two indexes, a value of zero corresponds to elastic isotropy, while any departure from zero corresponds to an extent of anisotropy. Similar to the ratio of the two extremes for Young’s modulus, the positive values of $A^{U}$ and $A^{L}$ for both LPSs at 0 K increase monotonically with increasing the pressure over the studied entire pressure range, showing the increase of their elastic anisotropies with pressure. At identical pressures the $A^{U}$($A^{L}$) value of the h LPS is always larger than that of the r one, indicating the stronger elastic anisotropy of the h LPS than the r one in the pressure range from 0 to 30 GPa.

### 3.4. Electronic Structures

The total and partial electronic densities of states (DOSs) of the LPSs h- and r-Al${}_{2}$Ti under different pressures at 0 K are calculated to fully understand the pressure effect on their structural and elastic properties. The obtained DOSs of both LPSs at zero pressure are plotted in Figure 12, in which except for Ti *d* DOS, the values of the other partial DOSs are enlarged by 10 times. The distribution curves of these total and partial DOSs are in good agreement with the results of the literature [25]. As can be clearly seen from Figure 12a, in the lower energy region between −10.0 and −4.0 eV, the total DOSs of the h LPS dominantly originate from the *s* states of Al, in the energy region between −4.0 and 0.0 eV, its total DOSs are determined by both Al *p* and Ti *d* states, and in the higher energy region between 0.0 and 3.0 eV, its total DOSs are mainly due to the contributions of Ti *d* electrons. Meanwhile, the DOS profiles for both Al *p* and Ti *d* are very similar in the energy range from −4.0 to 3.0 eV, reflecting the strong hybridization between these two orbits. Moreover, there is the observation of a pseudogap near the Femi level in the total DOS of the h LPS. The pseudogap also shows the appearance of the hybridization in the h one. These facts suggest that covalent bonds exist in the h LPS.

Comparing Figure 12b with Figure 12a in detail, one can see that the total and partial DOSs of the r LPS exhibit a similar feature with those of the h one. For the r LPS, the three energy ranges are from −10.0 to −4.1 eV, from −4.1 to 0.0 eV and from 0.0 to 1.8 eV, respectively. In each energy region, the contributions of various valence electrons to the total DOS of the r LPS are very similar to those of the h one. Thus, it can be concluded that there are the occurrences of the strong hybridization between Al *p* and Ti *d* in the r LPS, and the pseudogap near the Femi level in its total DOS, showing the existence of the covalent bonds in the r one. As described in Ref. [25], the Fermi energy falls in the descending part of the DOS for the h LPS, while it is situated directly in the pseudogap for the r one. It is generally accepted that a lower DOS value at the Fermi level corresponds to a higher stability of a crystalline structure [50]. The present DOS values at the Fermi level for the h (0.356 states/eV per atom) and r (0.338 states/eV per atom) LPSs at 0 GPa are in excellent agreement with those (0.357 and 0.335 states/eV per atom) of the literature [25]. It is suggested that at zero pressure the h LPS has a lower structural stability than the r one.

The obtained total DOSs of the LPS h- and r-Al${}_{2}$Ti at 0, 15 and 30 GPa are plotted in Figure 13, together with the partial DOSs of Al *s*, *p* and Ti *d* electrons. Clearly, the profiles of the total and partial DOSs for both LPSs are changed slightly with increasing the pressure. It is indicated that both LPSs are still structurally stable under pressure up to 30 GPa, and the types of their orbital hybridization are unchanged with increasing pressure. Owing to the presence of the pseudogap, the total DOSs of both LPSs at different pressures can be divided into one low energy bonding and the other high energy antibonding regions. It can be clearly seen from Figure 13a,d that as the pressure increases, the DOS peaks in the bonding region shift to the lower energy region and those in the antibonding region shift to the higher energy region. This situation results in the wider pseudogaps of both LPSs with pressure, meaning that they have more covalent bonding with pressure. Comparing Figure 13b with Figure 13c carefully, it can be found that the DOS profiles in the energy region between −4.0 and 3.0 eV for both Ti *d* and Al *p* orbits become more similar with increasing the pressure. A similar phenomenon can also be observed by carefully comparing Figure 13e with Figure 13f. These also show the stronger hybridization between Ti *d* and Al *p* for both LPSs with pressure. Consequently, both LPSs become more brittle with pressure. These are consistent with the above analysis of structural and elastic properties.

### 3.5. Thermodynamic Properties

Thermodynamic properties are of fundamental interest in condensed matter physics and material science. They can be derived from equilibrium achieved under high temperature and high pressure. Herein, the thermodynamic properties of h- and r-Al${}_{2}$Ti LPSs under pressure up to 30 GPa are investigated over a temperature range from 0 to 1700 K. The present variations of the equilibrium volumes with temperature for both LPSs under pressure up to 30 GPa are plotted in Figure 14. It can be clearly seen that the volumes of both LPSs at a given pressure can expand with rising the temperature, while their volumes at a given temperature can shrink with increasing the pressure. Meanwhile, the volumes of both LPSs under various pressures show a slight expansion up to around 100 K, and then a linear trend of sharp expansion at higher temperatures. The curves of the temperature dependent volumes under various pressures for either LPS are almost parallel to each other, and thus the volume difference of either LPS at two pressures is almost unchanged with rising the temperature. Comparing Figure 14a with Figure 14b carefully, it can be found that at identical pressure, the volume expansion ratio of the h LPS is same with that of the r one at low temperature (<500 K), and is slightly smaller than that of the r one at high temperature (>500 K). At zero temperature and pressure, the volumes of the h and r LPSs correspond to 191.60 and 385.80 Å${}^{3}$, which are well self-consistent with those (192.90 and 383.03 Å${}^{3}$) of the above structural properties.

The present variations of the isothermal bulk moduli with temperature for both LPSs under pressure up to 30 GPa are plotted in Figure 15. One can clearly see that the bulk moduli of both LPSs at a given pressure can decrease with rising the temperature, while their bulk moduli at a given temperature can increase with increasing the pressure. Similar to the changes of the volumes with temperature and pressure, the bulk moduli of both LPSs under various pressures show a slight decrease up to about 100 K and then a linear trend of sharp decrease at higher temperatures. The curves of the temperature dependent moduli under various pressures for either LPS are almost parallel to each other, and thus the modulus difference of either LPS at two pressures is almost unchanged with rising the temperature. These indicate the mechanical stabilities of both LPSs over the studied temperature and pressure ranges. At zero temperature and pressure, the bulk moduli of the h and r LPSs correspond to 106.78 and 106.56 GPa, which are also well self-consistent with those (107.91 and 108.42 GPa) of the above elastic properties.

From the temperature dependent equilibrium volumes of both LPSs at a given pressure, the volumetric thermal expansion coefficient $\beta_{V}$ as a function of temperature *T* can be determined by
(14)$$\begin{array}{c} \beta_{V}\left(T\right)=\frac{1}{V}{\left(\frac{\partial V}{\partial T}\right)}_{P}{|}_{V=V_{0}\left(T\right)}. \end{array}$$
where, $V_{0}$(*T*) is the equilibrium volume at *T*. The present variations of the volumetric thermal expansion coefficient with temperature for both LPSs under pressure up to 30 GPa are plotted in Figure 16. It is clear that as the temperature rises, the thermal expansion coefficients of either LPS at a given pressure dramatically increase at low temperature (<300 K) and then gradually tend to a linear growth at high temperature (>300 K), while the thermal expansion coefficients of either LPS at a given temperature decrease with increasing the pressure, and the higher the temperature, the faster the coefficients decrease. The effects of the pressure on the coefficients for either LPS are small at low temperature, and then increase with the rising temperature.

Heat capacity is one of the most essential thermodynamic properties of solids. The present variations of the heat capacity with temperature for the LPSs h- and r-Al${}_{2}$Ti at 0 and 30 GPa are plotted in Figure 17. In the figure, $C_{p}$ and $C_{v}$ denote the heat capacity at constant pressure and constant volume, respectively. At a given pressure, the difference between $C_{p}$ and $C_{v}$ for both LPSs can be calculated as $\beta_{V}^{2}BTV$. Clearly, the difference between $C_{p}$ and $C_{v}$ for either LPS at any pressure is very small in the temperature range from 0 to 1700 K. Meanwhile, the heat capacities $C_{p}$ and $C_{v}$ of either LPS increase with rising the temperature at identical pressure and decrease with increasing the pressure at identical temperature. The effects of the temperature on their heat capacities are much more significant than those of the pressure. At low temperature (300 K), the temperature dependent curves of $C_{p}$ and $C_{v}$ for both LPSs are similar to each other, which are proportional to $T^{3}$. At high temperature (>300 K), the $C_{p}$ curve of either LPS tend to be linear with temperature, while the $C_{v}$ curve gradually deviates from the $C_{p}$ one with temperature and approaches the Dulong–Petit limit.

To predict the temperature at which the h-Al${}_{2}$Ti LPS transforms to the r-Al${}_{2}$Ti one, the Gibbs free energy difference between both h- and r-Al${}_{2}$Ti LPSs were calculated in the temperature range from 0 to 1700 K and the pressure range from 0 to 30 GPa. The present variations of the Gibbs free energy difference between both LPSs at different pressures with temperature are plotted in Figure 18. The positive value denotes that the Gibbs free energy of the h LPS is higher and the superstructure is more unstable than the r one. Clearly, the Gibbs free energy difference between LPSs decreases gradually with rising the temperature at same pressure, while it decreases with increasing the pressure at same temperature, and the higher the temperature, the faster the Gibbs free energy difference decreases. The effects of the pressure on the Gibbs free energy difference for either LPS are small at low temperature, and then increase with the rising temperature. Specifically, the Gibbs free energy difference is equal to zero at 0 GPa when the temperature reaches 1399 K. When the pressure increases to 5 GPa, the transformation temperature decreases to 1335 K accordingly. Further increasing the pressure to 10, 15, 20, 25 and 30 GPa, the phase transitions between both h and r LPSs take place at 1289, 1252, 1222, 1195 and 1170 K, respectively. Comprehensively, the present results indicate that the h-Al${}_{2}$Ti LPS at various pressures can exist as a metastable phase in the temperature range from 0 to 1700 K and the phase transition temperature between both the h- and r-Al${}_{2}$Ti ones is 1399 K at zero pressure. This elucidates the long unclear and controversial thermal stability of the h-Al${}_{2}$Ti LPS.

## 4. Conclusions

The structural parameters, elastic properties, electronic structures and thermodynamic properties of h- and r-Al${}_{2}$Ti long-period superstructures under high pressure were studied by first-principles calculations in combination with quasi-harmonic approximation. The optimized structural parameters were in excellent agreement with the experimental and other theoretical values. These structural parameters showed the monotonic decrease with pressure. The hardest compression axis was the *a* for the h LPS and the *c* for the r LPS, while the easiest compression axis was the *c* for the h LPS and the *a* one for the r LPS. The volumes compressions of both LPSs were almost identical to each other. The single-crystal elastic constants $C_{ij}$ under various pressures were calculated by the strain-stress relationship. The elastic constants at zero pressure were highly consistent with the other theoretical results. According to these constants and phonon calculations, both LPSs were mechanically and dynamically stable in the pressure range from 0 to 30 GPa. All the elastic constants showed a linear increase with pressure. The polycrystalline bulk, shear and Young’s moduli, ratio of bulk to shear modulus, Poisson’s ratio and Cauchy pressures under various pressures were obtained in terms of the elastic constants which indicated that both LPSs had more covalent bonding and became more brittle with pressure. The directional dependence of bulk, shear and Young’s moduli at various pressures was also obtained in terms of the elastic constants, together with several anisotropy parameters. These showed that the strong anisotropy of both LPSs became more significant with pressure, and the anisotropy of the h LPS was stronger than that of the r one. Electronic structures of both LPSs at different pressures were determined to get some insight into the bonding characteristics and mechanical properties. Both LPSs had the wider pseudogaps and stronger hybridization between Ti *d* and Al *p* with pressure, which showed that they had more angular bonding with pressure. The obtained dependence of the volume, isothermal bulk modulus, thermal expansion coefficient, heat capacity and Gibbs free energy difference on temperature and pressure indicated that at identical pressure, the volume, thermal expansion coefficient and heat capacity increased with the rising temperature, while the isothermal bulk modulus and Gibbs free energy difference decreased with the rising temperature. Meanwhile, at identical temperature, the volume, thermal expansion coefficient, heat capacity and Gibbs free energy difference decreased with the increase of the pressure, while the isothermal bulk modulus increased with the increase of the pressure. These results can provide useful information for the further optimization design of high performance TiAl alloys, which shall promote the alloys for applications in aerospace, automotive and other industries.

## Figures and Tables

**Figure 1 materials-15-04236-f001:**
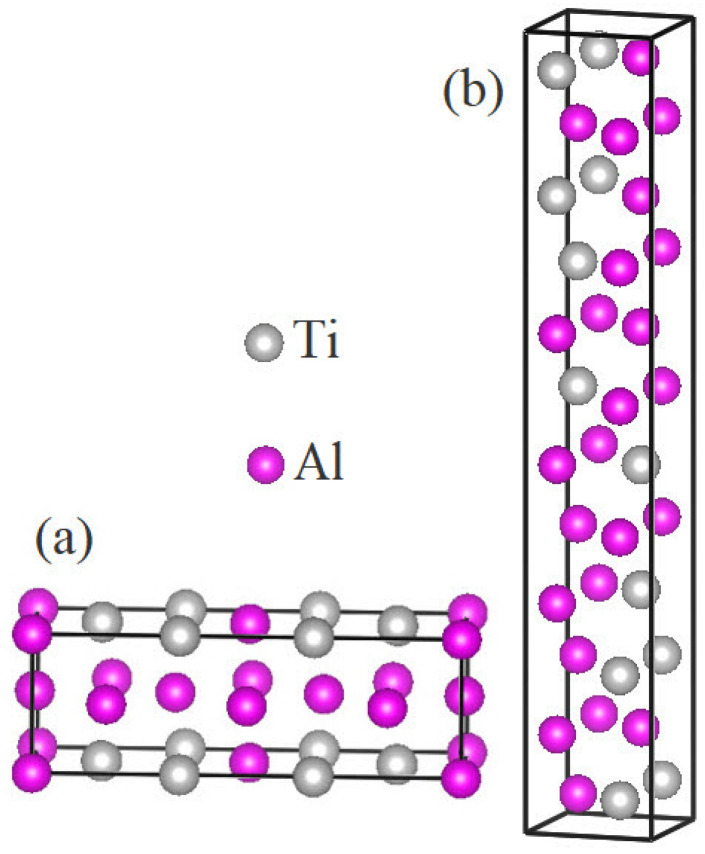
The unit cells of (**a**) h-Al${}_{2}$Ti and (**b**) r-Al${}_{2}$Ti.

**Figure 2 materials-15-04236-f002:**
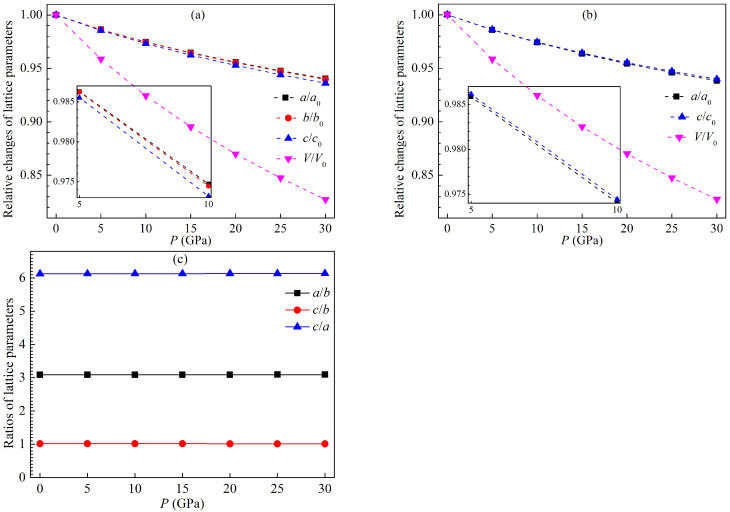
Relative changes of structural parameters $a/a_{0}$, $b/b_{0}$, $c/c_{0}$ and $V/V_{0}$ for the LPSs (**a**) h-Al${}_{2}$Ti and (**b**) r-Al${}_{2}$Ti at 0 K, and (**c**) the a/b and c/b of the h LPS and the c/a of the r one as a function of pressure.

**Figure 3 materials-15-04236-f003:**
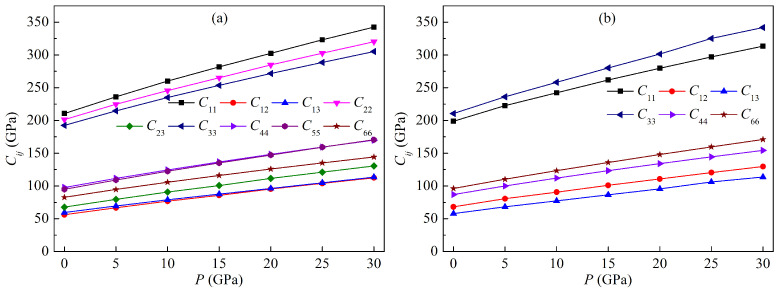
Elastic constants of the LPSs (**a**) h-Al${}_{2}$Ti and (**b**) r-Al${}_{2}$Ti as a function of pressure at 0 K.

**Figure 4 materials-15-04236-f004:**
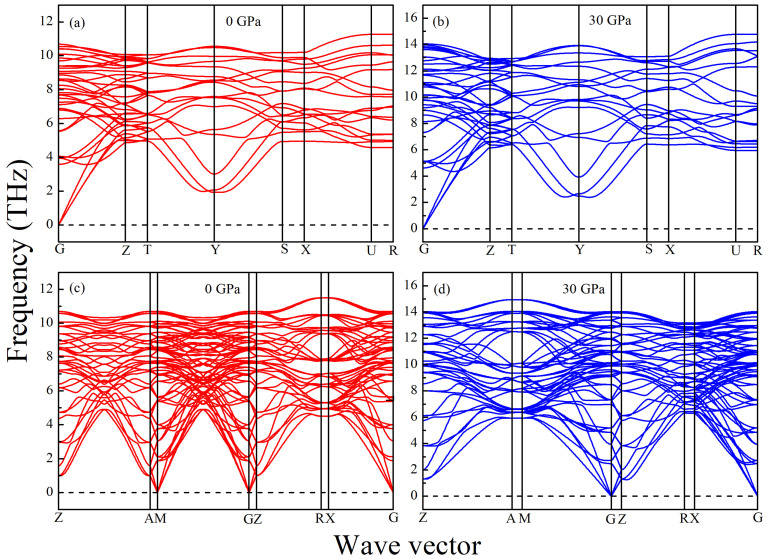
Phonon spectra of the LPSs (**a**,**b**) h-Al${}_{2}$Ti and (**c**,**d**) r-Al${}_{2}$Ti under 0 and 30 GPa at 0 K.

**Figure 5 materials-15-04236-f005:**
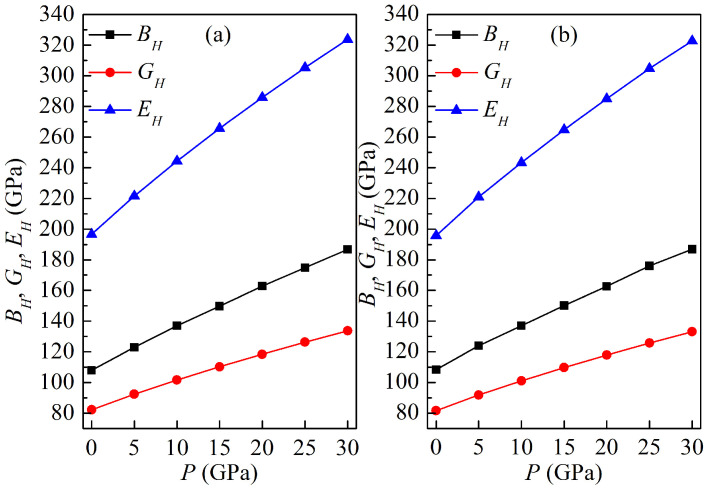
Elastic moduli of the LPSs (**a**) h-Al${}_{2}$Ti and (**b**) r-Al${}_{2}$Ti as a function of pressure at 0 K.

**Figure 6 materials-15-04236-f006:**
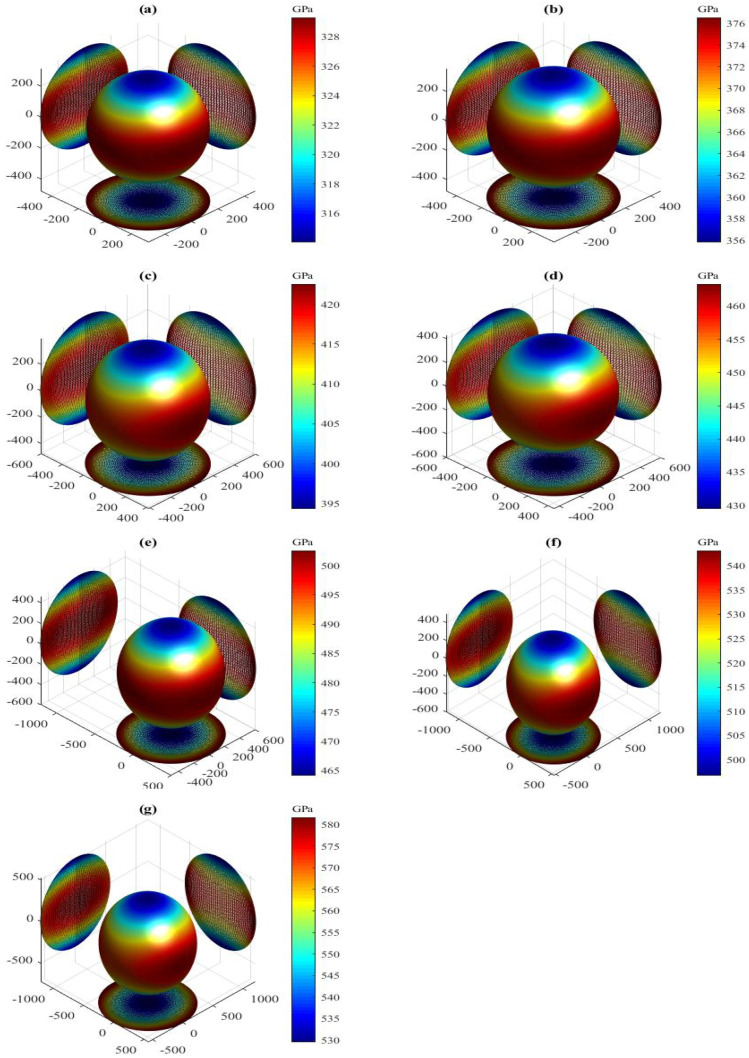
Directional dependence of bulk modulus for the LPS h-Al${}_{2}$Ti under pressure (**a**) 0 GPa, (**b**) 5 GPa, (**c**) 10 GPa, (**d**) 15 GPa (**e**) 20 GPa, (**f**) 25 GPa and (**g**) 30 GPa at 0 K.

**Figure 7 materials-15-04236-f007:**
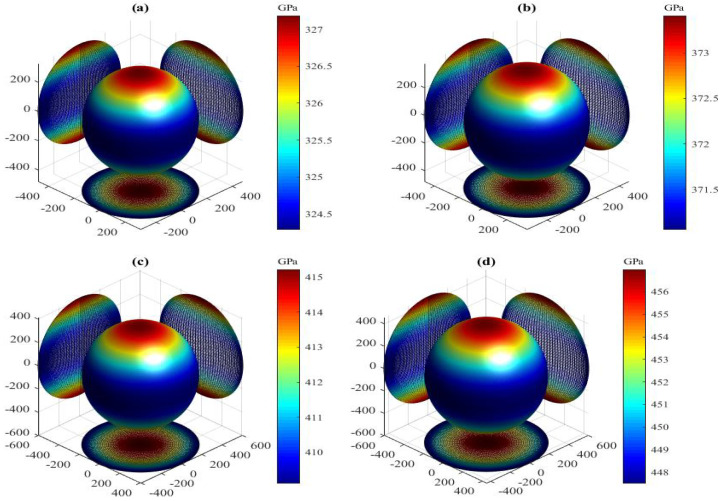
Directional dependence of bulk modulus for the LPS r-Al${}_{2}$Ti under pressure (**a**) 0 GPa, (**b**) 5 GPa, (**c**) 10 GPa, (**d**) 15 GPa (**e**) 20 GPa, (**f**) 25 GPa and (**g**) 30 GPa at 0 K.

**Figure 8 materials-15-04236-f008:**
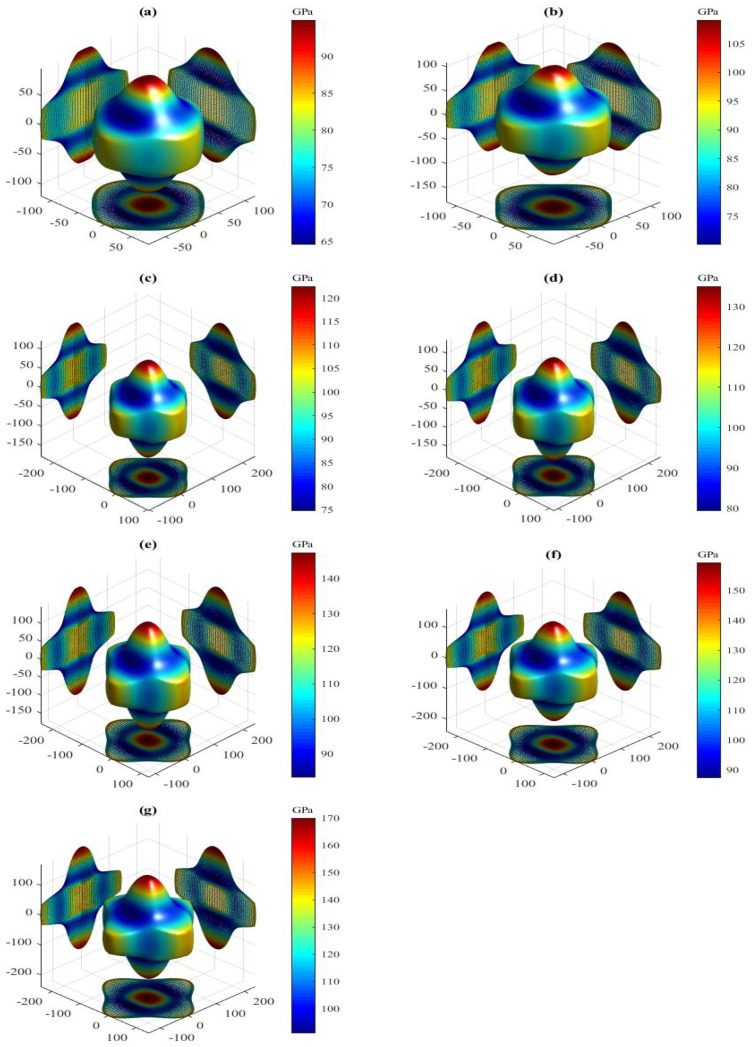
Directional dependence of shear modulus for the LPS h-Al${}_{2}$Ti under pressure (**a**) 0 GPa, (**b**) 5 GPa, (**c**) 10 GPa, (**d**) 15 GPa (**e**) 20 GPa, (**f**) 25 GPa and (**g**) 30 GPa at 0 K.

**Figure 9 materials-15-04236-f009:**
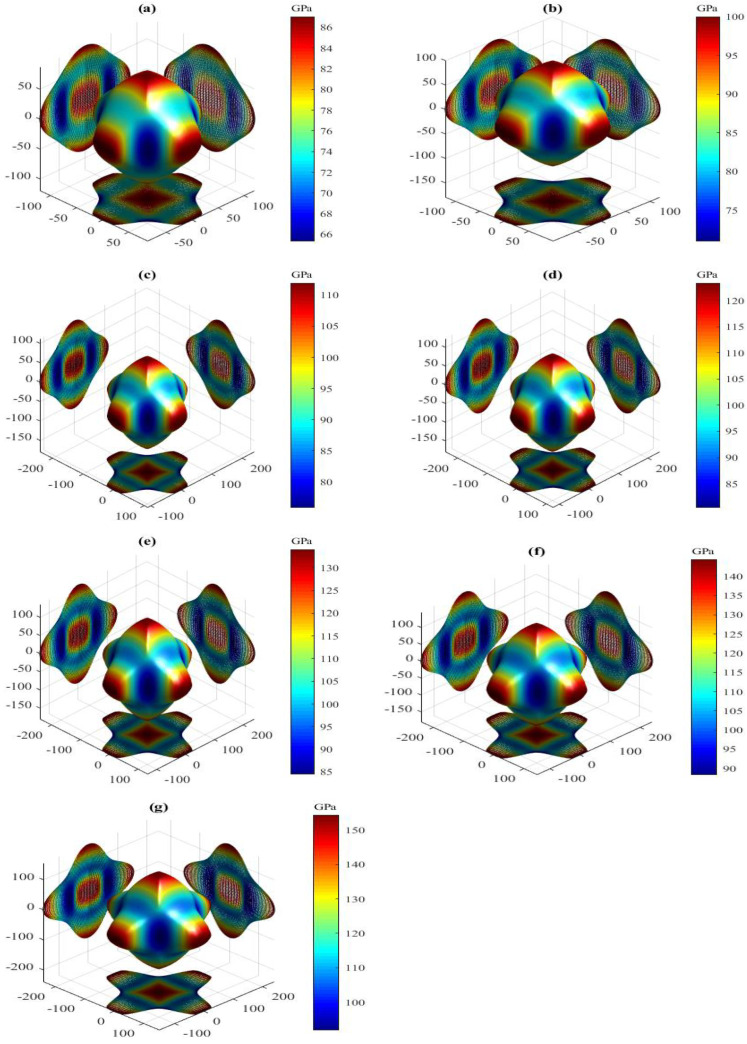
Directional dependence of shear modulus for the LPS r-Al${}_{2}$Ti under pressure (**a**) 0 GPa, (**b**) 5 GPa, (**c**) 10 GPa, (**d**) 15 GPa (**e**) 20 GPa, (**f**) 25 GPa and (**g**) 30 GPa at 0 K.

**Figure 10 materials-15-04236-f010:**
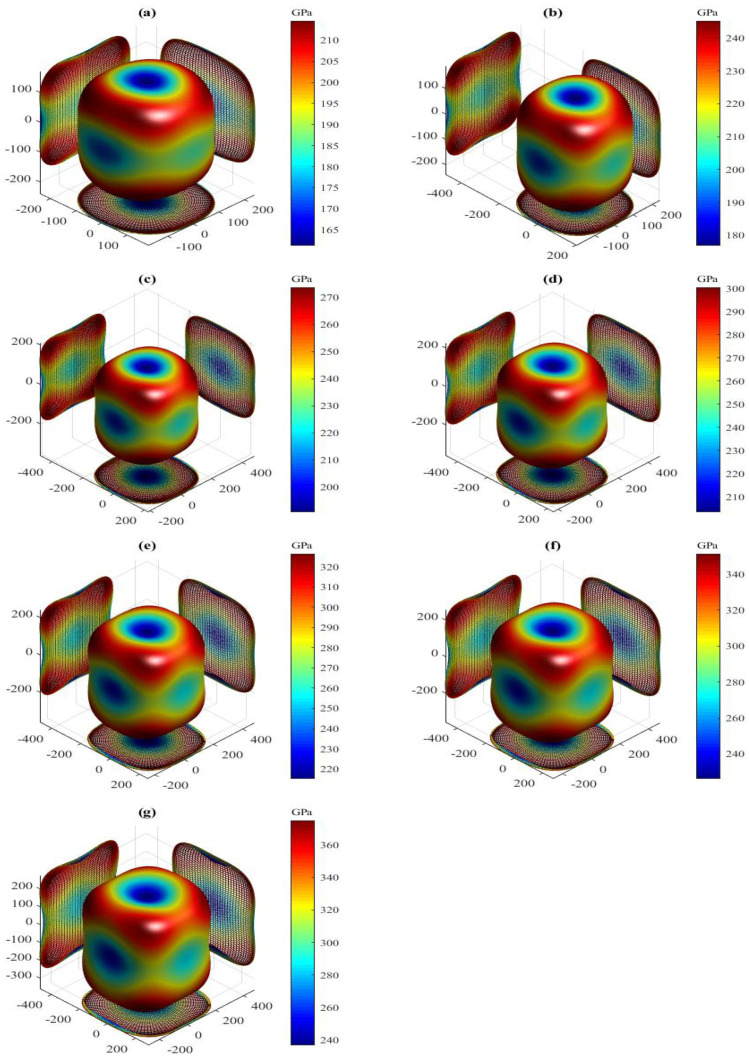
Directional dependence of Young’s modulus for the LPS h-Al${}_{2}$Ti under pressure (**a**) 0 GPa, (**b**) 5 GPa, (**c**) 10 GPa, (**d**) 15 GPa (**e**) 20 GPa, (**f**) 25 GPa and (**g**) 30 GPa at 0 K.

**Figure 11 materials-15-04236-f011:**
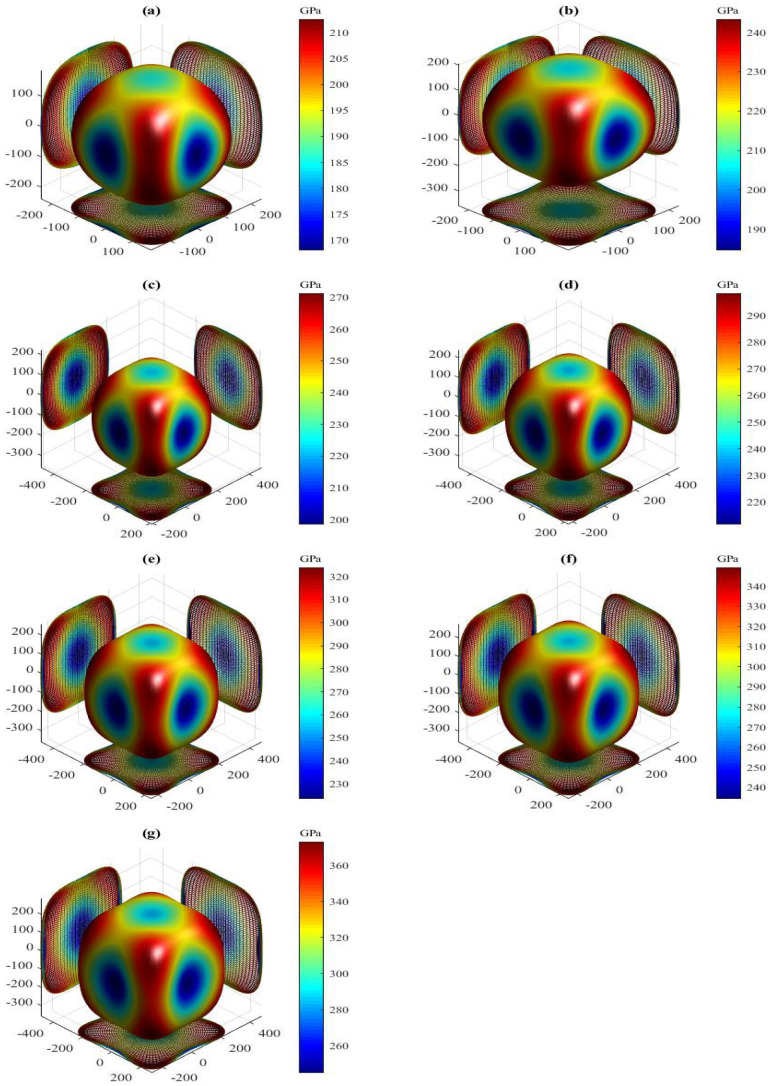
Directional dependence of Young’s modulus for the LPS r-Al${}_{2}$Ti under pressure (**a**) 0 GPa, (**b**) 5 GPa, (**c**) 10 GPa, (**d**) 15 GPa (**e**) 20 GPa, (**f**) 25 GPa and (**g**) 30 GPa at 0 K.

**Figure 12 materials-15-04236-f012:**
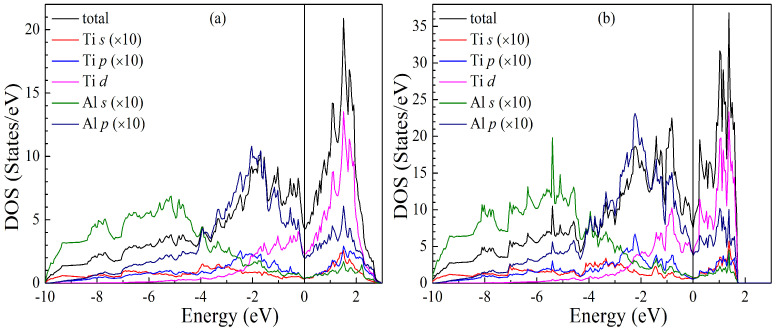
Calculated total and partial densities of states (DOSs) for the LPS (**a**) h- and (**b**) r-Al${}_{2}$Ti at 0 GPa and 0 K. The Fermi level is shift to 0 eV.

**Figure 13 materials-15-04236-f013:**
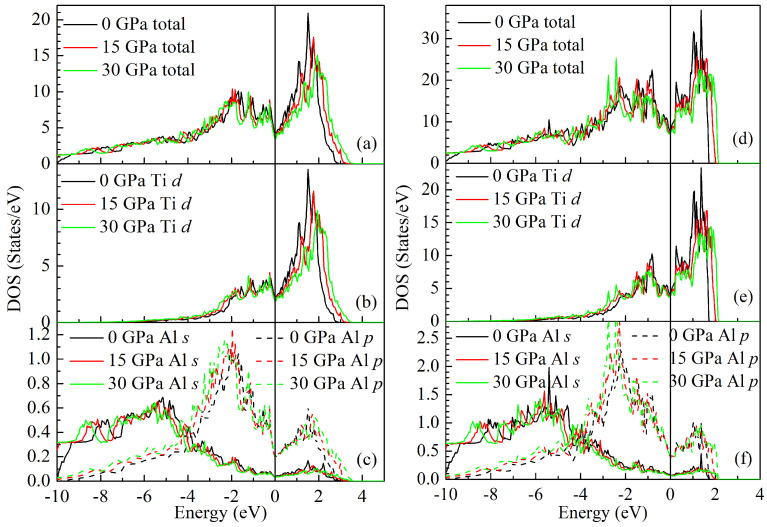
Calculated total and partial densities of states (DOSs) for the LPS h- and r-Al${}_{2}$Ti at the pressures of 0, 15 and 30 GPa and the temperature of 0 K: (**a**,**d)** for total DOS, (**b**,**e**) for the DOS of Ti *d* orbital, (**c**,**f**) for the DOSs of Al *s* and *p* orbitals. The Fermi level is shift to 0 eV.

**Figure 14 materials-15-04236-f014:**
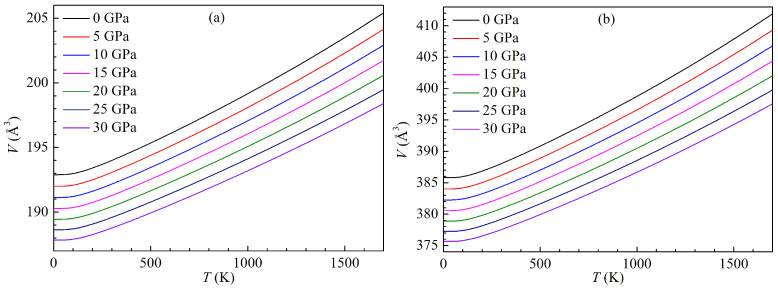
Temperature dependence of volume *V* for the LPSs (**a**) h- and (**b**) r-Al${}_{2}$Ti under pressure up to 30 GPa.

**Figure 15 materials-15-04236-f015:**
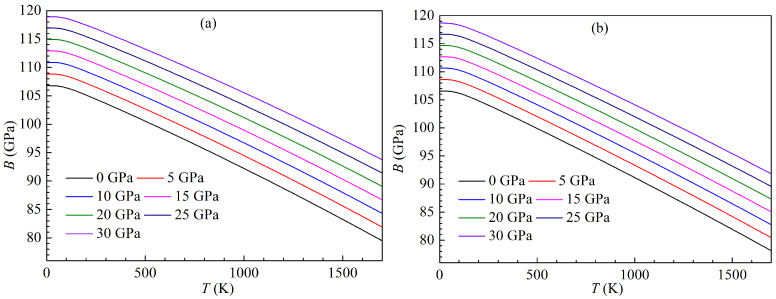
Temperature dependence of bulk modulus *B* for the LPSs (**a**) h- and (**b**) r-Al${}_{2}$Ti under pressure up to 30 GPa.

**Figure 16 materials-15-04236-f016:**
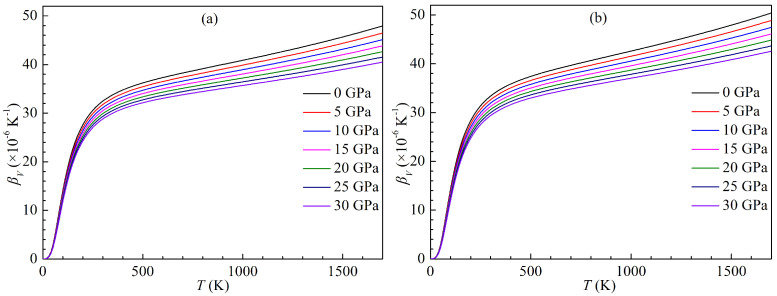
Temperature dependence of thermal expansion coefficient $\beta_{V}$ for the LPSs (**a**) h- and (**b**) r-Al${}_{2}$Ti under pressure up to 30 GPa.

**Figure 17 materials-15-04236-f017:**
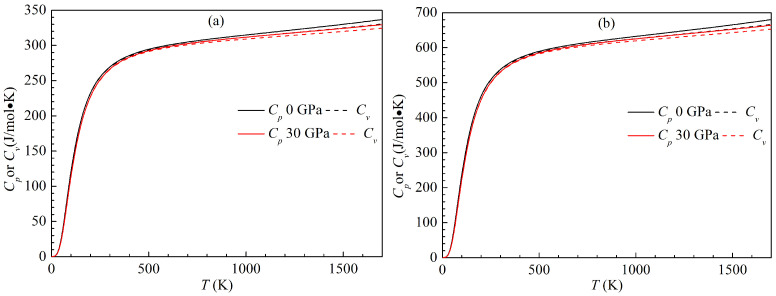
Temperature dependence of heat capacity at constant pressure $C_{p}$ and volume $C_{v}$ for the LPSs (**a**) h- and (**b**) r-Al${}_{2}$Ti at 0 GPa and 30 GPa.

**Figure 18 materials-15-04236-f018:**
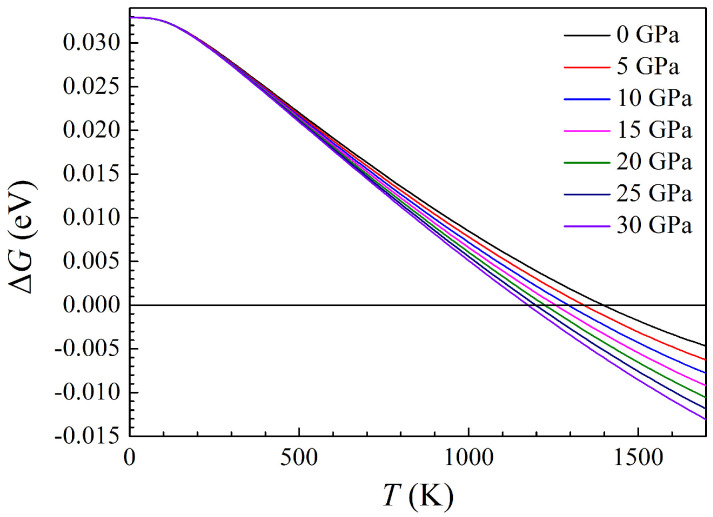
Temperature dependence of the Gibbs free energy difference (ΔG) between the h- and the r-Al${}_{2}$Ti LPSs under pressure up to 30 GPa. The positive value shows that the h LPS is more stable than the r one.

**Table 1 materials-15-04236-t001:** Calculated bulk properties of the LPSs h- and r-Al${}_{2}$Ti at 0 GPa and 0 K, togerther with available experimental and theoretical data.

LPS	h						r					
	Present	[21]	[25]	[26]	[24]	[44]	Present	[22]	[25]	[26]	[24]	[44]
$a_{0}$	12.158	12.094	12.141	12.144	12.161	12.157	3.969	3.971	3.966	3.971	3.966	3.969
$b_{0}$	3.932	3.959	3.931	3.923	3.932	3.932	3.969	3.971	3.966	3.971	3.966	3.969
$c_{0}$	4.001	4.032	4.002	4.008	4.002	4.004	24.316	24.313	24.307	24.310	24.321	24.284
$V_{0}$	191.60	193.04	191.01	190.90	191.36	191.38	383.03	383.39	382.33	383.35	382.50	382.52
$C_{11}$	210.57		208.70			209	198.81		198.20			201
$C_{12}$	56.17		39.87			56	68.17		71.67			67
$C_{13}$	59.48		46.76			62	57.81		61.41			58
$C_{22}$	201.41		193.82			201	198.81		198.20			201
$C_{23}$	67.74		74.38			70	57.81		61.41			58
$C_{33}$	192.50		187.25			189	210.52		208.76			209
$C_{44}$	97.75		97.05			98	87.05		83.58			87
$C_{55}$	94.87		92.35			96	87.05		83.58			87
$C_{66}$	82.70		76.76			80	96.16		95.44			97
$B_{H}$	107.91		101.275			108	108.42		110.460			109
$G_{H}$	82.17		80.449			81	81.61		79.137			82
$E_{H}$	196.61		190.820				195.72		191.644			
$\nu_{H}$	0.1963		0.186			0.20	0.1991		0.211			0.20

**Table 2 materials-15-04236-t002:** Optimized lattice parameters *a*, *b* and *c* (in Å) and volumes of unit cell *V* (in Å${}^{3}$) for the LPSs h- and r-Al${}_{2}$Ti under pressure up to 30 GPa at 0 K.

LPS	*P*	*a*	*b*	*c*	*V*
	0	12.158	3.932	4.001	191.60
	5	11.991	3.878	3.950	183.67
	10	11.850	3.832	3.900	177.11
h	15	11.728	3.792	3.857	171.52
	20	11.619	3.756	3.819	166.66
	25	11.522	3.725	3.783	162.36
	30	11.433	3.695	3.752	158.50
	0	3.969	3.969	24.316	383.03
	5	3.913	3.913	23.979	367.11
	10	3.866	3.866	23.694	354.11
r	15	3.824	3.824	23.446	342.93
	20	3.788	3.788	23.228	333.23
	25	3.754	3.754	23.030	324.62
	30	3.724	3.724	22.851	316.92

**Table 3 materials-15-04236-t003:** Calculated elastic constants (in GPa) for the LPSs h- and r-Al${}_{2}$Ti under pressure up to 30 GPa at 0 K.

LPS	*P*	$C_{11}$	$C_{12}$	$C_{13}$	$C_{22}$	$C_{23}$	$C_{33}$	$C_{44}$	$C_{55}$	$C_{66}$
	0	210.57	56.17	59.48	201.41	67.74	192.50	97.75	94.87	82.70
	5	236.08	66.71	69.42	224.63	79.55	214.71	111.67	109.11	94.65
	10	260.21	76.80	78.95	245.62	90.66	235.02	124.53	122.46	105.73
h	15	281.86	85.74	87.60	265.17	100.72	253.59	136.78	135.13	116.06
	20	302.50	95.28	96.26	284.88	111.51	271.65	148.32	147.29	125.87
	25	323.03	103.94	104.95	302.72	121.01	288.56	159.42	159.27	135.24
	30	342.50	112.49	113.53	320.08	130.57	305.22	170.06	170.51	144.22
	0	198.81	68.17	57.81	198.81	57.81	210.52	87.05	87.05	96.16
	5	222.59	80.63	68.27	222.59	68.27	236.01	100.01	100.01	110.26
	10	242.32	90.63	77.30	242.32	77.30	258.31	111.95	111.95	123.34
r	15	261.75	100.99	86.56	261.75	86.56	280.20	123.37	123.37	135.93
	20	279.63	110.72	95.57	279.63	95.57	301.32	134.04	134.04	148.01
	25	297.10	120.54	106.13	297.10	106.13	325.17	144.42	144.42	159.73
	30	313.45	129.72	113.67	313.45	113.67	341.80	154.36	154.36	171.03

**Table 4 materials-15-04236-t004:** Calculated elastic moduli and Cauchy pressures (in GPa) for the LPSs h- and r-Al${}_{2}$Ti under pressure up to 30 GPa at 0 K.

LPS	*P*	$B_{H}$	$G_{H}$	$E_{H}$	$B_{H}/G_{H}$	$\nu_{H}$	${\mathrm{CP}}_{1}$	${\mathrm{CP}}_{2}$	${\mathrm{CP}}_{3}$
	0	107.91	82.17	196.61	1.3132	0.1963	−26.53	−35.39	−30.02
	5	122.96	92.32	221.51	1.3319	0.1997	−27.94	−39.69	−32.12
	10	137.05	101.60	244.41	1.3489	0.2028	−28.93	−43.51	−33.87
h	15	149.83	110.29	265.67	1.3585	0.2045	−30.32	−47.52	−36.06
	20	162.75	118.42	285.92	1.3744	0.2072	−30.60	−51.03	−36.80
	25	174.85	126.27	305.31	1.3848	0.2090	−31.30	−54.32	−38.42
	30	186.72	133.70	323.81	1.3966	0.2110	−31.73	−56.99	−39.49
	0	108.42	81.61	195.72	1.3284	0.1991	−27.99	−29.24	−29.24
	5	123.95	91.86	220.99	1.3492	0.2028	−29.63	−31.74	−31.74
	10	137.04	101.05	243.35	1.3561	0.2041	−32.71	−34.65	−34.65
r	15	150.21	109.79	264.84	1.3682	0.2061	−34.95	−36.81	−36.81
	20	162.69	117.91	284.91	1.3798	0.2081	−37.29	−38.47	−38.47
	25	176.08	125.74	304.69	1.4003	0.2116	−39.20	−38.29	−38.29
	30	186.96	133.10	322.71	1.4047	0.2123	−41.31	−40.69	−40.69

**Table 5 materials-15-04236-t005:** Calculated external ($B_{max}$, $B_{min}$) and linear ($B_{a}$, $B_{b}$, $B_{c}$) bulk moduli (in GPa) and relative ratios ($B_{max}/B_{min}$, $B_{Ba}$, $B_{Bc}$) for the LPSs h- and r-Al${}_{2}$Ti under pressure up to 30 GPa at 0 K.

LPS	*P*	$B_{\max}$	$B_{\min}$	$B_{\max}/B_{\min}$	$B_{a}$	$B_{b}$	$B_{c}$	$B_{\mathrm{Ba}}$	$B_{\mathrm{Bc}}$
	0	329.30	314.05	1.0486	329.30	328.16	314.05	1.0035	0.9570
	5	376.56	355.83	1.0582	376.56	374.84	355.83	1.0046	0.9493
	10	422.54	394.33	1.0715	422.54	417.49	394.33	1.0121	0.9445
h	15	463.33	429.47	1.0788	463.33	456.84	429.47	1.0142	0.9401
	20	502.57	464.14	1.0828	502.57	499.64	464.14	1.0058	0.9289
	25	543.19	496.73	1.0935	543.19	535.74	496.73	1.0139	0.9272
	30	581.70	529.55	1.0984	581.70	571.53	529.55	1.0178	0.9266
	0	327.19	324.28	1.0089	324.28	324.28	327.19	1.0000	1.0089
	5	373.41	371.05	1.0064	371.05	371.05	373.41	1.0000	1.0064
	10	415.22	409.11	1.0149	409.11	409.11	415.22	1.0000	1.0149
r	15	456.99	447.49	1.0212	447.49	447.49	456.99	1.0000	1.0212
	20	498.73	482.89	1.0328	482.89	482.89	498.73	1.0000	1.0328
	25	551.55	517.15	1.0665	517.15	517.15	551.55	1.0000	1.0665
	30	582.12	550.71	1.0570	550.71	550.71	582.12	1.0000	1.0570

**Table 6 materials-15-04236-t006:** Calculated maximal ($G_{max}$) and minimal ($G_{min}$) values of shear modulus, the ratio of the two extremes ($G_{max}/G_{min}$), and shear anisotropic factors ($A_{\left\{100\right\}}$, $A_{\left\{010\right\}}$, $A_{\left\{001\right\}}$) for the LPSs h- and r-Al${}_{2}$Ti under pressure up to 30 GPa at 0 K.

LPS	*P*	$G_{\max}$	$G_{\min}$	$G_{\max}/G_{\min}$	$A_{\left\{100\right\}}$	$A_{\left\{010\right\}}$	$A_{\left\{001\right\}}$
	0	94.87	64.54	1.4700	1.3763	1.4684	1.1040
	5	109.11	70.00	1.5588	1.4319	1.5574	1.1567
	10	122.46	74.77	1.6378	1.4767	1.6366	1.2007
h	15	135.13	79.27	1.7046	1.5187	1.7033	1.2362
	20	147.29	83.31	1.7679	1.5546	1.7666	1.2688
	25	159.27	87.25	1.8255	1.5875	1.8241	1.2946
	30	170.06	90.97	1.8694	1.6171	1.8730	1.3182
	0	87.05	65.33	1.3326	1.1856	1.1856	1.4722
	5	100.01	70.99	1.4088	1.2421	1.2421	1.5535
	10	111.95	75.86	1.4757	1.2941	1.2941	1.6261
r	15	123.37	80.39	1.5345	1.3379	1.3379	1.6912
	20	134.04	84.47	1.5868	1.3755	1.3755	1.7525
	25	144.42	88.30	1.6355	1.4090	1.4090	1.8093
	30	154.36	91.88	1.6800	1.4430	1.4430	1.8618

**Table 7 materials-15-04236-t007:** Calculated maximal ($E_{max}$) and minimal ($E_{min}$) values of Young’s modulus, the ratio of the two extremes ($E_{max}/E_{min}$), and anisotropic factors ($A^{U}$, $A^{L}$) for the LPSs h- and r-Al${}_{2}$Ti under pressure up to 30 GPa at 0 K.

LPS	*P*	$E_{\max}$	$E_{\min}$	$E_{\max}/E_{\min}$	$A^{U}$	$A^{L}$
	0	214.60	161.27	1.3307	0.1192	0.0526
	5	245.20	176.84	1.3865	0.1562	0.0687
	10	273.71	190.71	1.4352	0.1915	0.0839
h	15	300.53	203.41	1.4774	0.2238	0.0977
	20	326.40	215.15	1.5171	0.2550	0.1110
	25	351.22	226.20	1.5527	0.2838	0.1232
	30	375.02	236.86	1.5833	0.3097	0.1341
	0	212.65	168.00	1.2658	0.0904	0.0401
	5	243.40	184.59	1.3186	0.1242	0.0549
	10	271.31	198.44	1.3672	0.1583	0.0697
r	15	298.41	211.59	1.4104	0.1905	0.0836
	20	324.04	223.43	1.4503	0.2215	0.0969
	25	349.30	234.40	1.4902	0.2516	0.1096
	30	373.00	245.04	1.5222	0.2805	0.1220

## Data Availability

Not applicable.

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
