# Peer review of "Theoretical Study on the Structural, Elastic, Electronic and Thermodynamic Properties of Long-Period Superstructures h- and r-Al2Ti under High Pressure"

_materials, 2022, doi:10.3390/ma15124236_

Round 1

Reviewer 1 Report

The authors have investigated structural, elastic, electronic and thermodynamic properties of h- and r-Al2Ti long- 1 period superstructures under pressure up to 30 GPa using density functional theory. The analysis is robust and wide-ranging and posses technical merit. The authors have carried out every effort to portray results in a detailed manner. This reviewer has following comments to further improve the paper:

1. The authors have given the motivation that "there is lack of knowledge of the structural, electronic, elastic and thermodynamics properties of h- 46 and r-Al2Ti LPSs under high pressure". It will be good if the authors add the importance of this missing knowledge thus further improving their motivation of this work.

2. The results given are very comprehensive, however since these are theoretical results, authors have compared them with available studies, however it make the reader very difficult to get a hand of the comparisons. It is therefore suggested to make a comprehensive table where the authors can put all their results and compare with available studies.

3. There are a few typos which demand authors to have a thorough review of the paper and remove any typos. (eg: hgigh in the last paragraph of Introduction and Fog. ?? in the first paragraph of Results and Discussion).

Author Response

Dear reviewer,

We would like to thank the comments from you, which much helpful for revision of our manuscript (#materials-1746049). We revised our manuscript accordingly and the corrections are listed below:

  1. The authors have given the motivation that "there is lack of knowledge of the structural, electronic, elastic and thermodynamics properties of h- and r-Al2Ti LPSs under high pressure". It will be good if the authors add the importance of this missing knowledge thus further improving their motivation of this work.

Thank you very much for your suggestion. We have added the importance of the missing knowledge in the revised version by using the states “It is known that the electronic, elastic and thermodynamic properties of intermetallic compounds under pressure are vital to the design and development of novel materials for structural applications. These properties are determined by the crystal structures.”

  1. The results given are very comprehensive, however since these are theoretical results, authors have compared them with available studies, however it make the reader very difficult to get a hand of the comparisons. It is therefore suggested to make a comprehensive table where the authors can put all their results and compare with available studies.

Thank you very much for your suggestion. We have made a comprehensive table where the authors can put all their results and compare with available studies, as seen in the section “3.1. Ground-state bulk properties” in the revised version.

  1. There are a few typos which demand authors to have a thorough review of the paper and remove any typos. (eg: hgigh in the last paragraph of Introduction and Fog. ?? in the first paragraph of Results and Discussion).

Thank you very much for your suggestion. We have corrected these typos in the revised version.

   Thank you again for your positive comments and valuable suggestions to improve the quality of our manuscript.

Best wishes.

Yufeng Wen

Jinggangshan University

Reviewer 2 Report

The article Theoretical study on the structural, elastic, electronic and thermodynamic properties of long-period superstructures h- and r-Al2Ti under high pressure is devoted to the study of structural, electronic and thermodynamic properties of h- and r-Al2Ti superstructures. Undoubtedly, the results presented by the authors are of high scientific novelty and practical significance, and are also promising for practical research. In general, the presented results of the study can be accepted for publication after the authors provide answers to all the questions raised by the reviewer during the reading of the article.

1. In the abstract, the authors need to more clearly state the purpose and relevance of this work.

2. The authors should explain what could be the reason for such a strong difference in the crystal lattice parameters, including the differences between the results of this work and the literature data.

3. Variations in the structural parameters depending on the pressure indicate the absence of anisotropy in the change in the crystal lattice parameters; the authors should explain what this is due to.

4. Figures 6 and 7 require further clarification in interpretation.

5. Technical remarks should include the need to provide measurement errors in the parameters of the crystal lattice in view of small changes in this value from pressure.

6. In conclusion, the authors should clearly indicate the prospects for further studies of these samples.

Author Response

Dear reviewer,

We would like to thank the comments from you, which much helpful for revision of our manuscript (#materials-1746049). We revised our manuscript accordingly and the corrections are listed below:

  1. In the abstract, the authors need to more clearly state the purpose and relevance of this work.

Thank you very much for your suggestion. In the abstract of the revised version, we have more clearly stated the purpose and relevance of this work by adding “The formations of long-period superstructures strongly influence the properties of Al-rich L10-TiAl intermetallic alloys. To soundly understand the role of the superstructures in the alloys, fundamentals about them have to be known”.

  1. The authors should explain what could be the reason for such a strong difference in the crystal lattice parameters, including the differences between the results of this work and the literature data.

Thank you very much for your suggestion. We have made a comprehensive table where the authors can put all their results and compare with available studies, as seen in the section “3.1. Ground-state bulk properties” in the revised version. There is a weaker difference (<1%) in the crystal lattice parameters, including the difference between the results of this work and the literature data.

  1. Variations in the structural parameters depending on the pressure indicate the absence of anisotropy in the change in the crystal lattice parameters; the authors should explain what this is due to.

Thank you very much for your suggestion. We have explained the absence of anisotropy in the change in the crystal lattice parameters by adding figure 2(c) and “One can also see from Fig. 2(c) that the ratios of a/b and c/b for the h PLS and c/a for the r one are all almost unchanged with increasing the pressure, meaning that both LPSs have a pressure isotropic structure.” in the revised version.

  1. Figures 6 and 7 require further clarification in interpretation.

Thank you very much for your suggestion. Further clarification in interpretation has been added for Figures 6 and 7 by the states “For these three-dimensional (3D) representation surfaces, the size of the bulk modulus is denoted by the length of the radius vector in arbitrary crystallographic directions and different colors. The 3D surface of an isotropic system must be a spherical shape with a color, and the nonsperical shape with different colors indicates the degree of anisotropy.” in the revised version.

  1. Technical remarks should include the need to provide measurement errors in the parameters of the crystal lattice in view of small changes in this value from pressure.

Thank you very much for your suggestion. We have added a partial enlargement subplot in Figs. 2(a) and (b) in the revised version to provide measurement errors in the parameters of the crystal lattice in view of small changes in this value from pressure.

  1. In conclusion, the authors should clearly indicate the prospects for further studies of these samples.

Thank you very much for your suggestion. We have clearly indicated the the prospects for further studies of these samples, by the state “These results can provide useful information for the further optimization design of high performance TiAl alloys, which shall promote the alloys for applications in aerospace, automotive and other industries.” in the revised version.

   Thank you again for your positive comments and valuable suggestions to improve the quality of our manuscript.

Best wishes.

Yufeng Wen

Jinggangshan University

Round 2

Reviewer 2 Report

The authors answered all the questions of the reviewer, the article can be accepted for publication.